# Is HIV Pre-Exposure Prophylaxis among Men Who Have Sex with Men Effective in a Real-World Setting? Experience with One-On-One Counseling and Support in a Sexual Health Center in Paris, 2018–2020

**DOI:** 10.3390/ijerph192114295

**Published:** 2022-11-01

**Authors:** Bérenger Thomas, Prescillia Piron, Elise de La Rochebrochard, Christophe Segouin, Pénélope Troude

**Affiliations:** 1Department of Public Health, University Hospital Lariboisière-Fernand-Widal, AP-HP, 75010 Paris, France; 2Free Sexual Health Center, University Hospital Lariboisière-Fernand-Widal, AP-HP, 75010 Paris, France; 3Institut National d’Etudes Démographiques (INED), 93300 Aubervilliers, France; 4CESP U1018, Inserm, UVSQ, Université Paris-Saclay, 94800 Villejuif, France

**Keywords:** HIV prevention, pre-exposure prophylaxis, MSM, retention in care, sexually transmitted infections

## Abstract

HIV pre-exposure prophylaxis (PrEP) is highly effective but depends on patients’ care engagement, which is often mediocre and poorly measured in real-world settings. This study aimed to assess the effectiveness of a PrEP program in a sexual health center that included accompanying measures to improve engagement. A retrospective observational study was conducted. All men who have sex with men (MSM) who initiated PrEP for the first time between 1 August 2018 and 30 June 2019 in the Fernand-Widal sexual health center, Paris, France, were included. Among the 125 MSM who initiated PrEP, the median age was 33 and most had only male partners. At initiation, 58% were considered at very high risk of HIV infection, mainly due to a history of post-exposure prophylaxis. During the first year, patients attended a median of three visits (Q1–Q3, 2–4). At 12 months, 96% (95% CI, 92.6 to 99.4) had a successful PrEP course, assessed by a novel metric. These results highlight the possibility of achieving a high PrEP success ratio among MSM in a real-world setting. The accompanying measures and one-on-one counseling by a trained counselor could explain the effectiveness of this PrEP program.

## 1. Introduction

To prevent human immunodeficiency virus (HIV) transmission, a combined prevention strategy has progressively emerged including treatment as prevention (TasP), post-exposure prophylaxis (PEP) and recently pre-exposure prophylaxis (PrEP). Using an antiretroviral combination, PrEP is taken before exposure to risk in order to prevent HIV infection. Its efficacy has been shown in several trials, especially among men who have sex with men (MSM), but it is particularly dependent on patient adherence [1,2]. To improve adherence and monitor other sexually transmitted infections (STIs) as well as possible side effects, follow-up guidelines recommend quarterly visits including clinical examination and laboratory tests. Moreover, to facilitate follow-up and broaden access to PrEP, several countries are diversifying care providers, notably including primary care providers [3].

However, the first studies in a real-world setting showed mediocre care engagement and encountered difficulties in adequately measuring the quality of follow-up with complex and variable courses of care [4]. Retention in care, usually assessed for HIV curative treatment, is largely used but does not seem optimal for preventive treatment. Unlike curative treatments, PrEP is not a life-long treatment, and the appropriateness of prescription must be reassessed over time depending on patients’ risk levels. Follow-up measures should therefore provide the possibility for patients to stop PrEP if changes in their sexual practices justify it. Moreover, these measures should take into account that patients may pursue their follow-up partially or totally in other structures, including primary care providers. Therefore, other metrics have been developed such as PrEP persistence, visit constancy or success ratio, including key parameters such as level of HIV infection risk and occurrence of HIV seroconversion [5,6,7]. Rather than evaluating the percentage of patients still followed in the center, these scores take account of the specific characteristics of this prevention strategy.

To enhance follow-up quality and care engagement, we developed a reinforced PrEP program in our sexual health center. Patients benefit from accompanying measures throughout their follow-up, relying notably on an identified and easily accessible referent counselor. This trained paramedic counselor in sexual health works in collaboration with all professionals in the center and is a bridge between care and administrative procedures. He provides support in organizing appointments with reminders, as well as phone and email follow-ups to answer daily questions. One-on-one sessions of therapeutic support and counseling are also offered at initiation and during follow-ups to assess risk practices, sexual well-being and improve PrEP understanding and adherence.

This study aimed to assess the effectiveness of such a PrEP program among MSM.

## 2. Materials and Methods

We conducted a retrospective observational study among PrEP users followed at the Fernand-Widal sexual health center, Paris. All MSM who initiated PrEP for the first time between 1 August 2018 and 30 June 2019 were included. Baseline characteristics and one-year follow-up data were collected using two complementary sources: the consultation database and patient records.

Routinely collected data included patients’ sociodemographic characteristics (age, nationality, place of birth and residence, work status), sexual practices (partners’ gender, history of STIs, PEP and chemsex practice in the last 12 months) and STIs at initiation based on systematical biological tests performed before initiation. Tests included serologies on blood samples for HIV and syphilis, and nucleic acid amplification testing on urine samples, pharyngeal and rectal self-swabs for chlamydia and gonococci. The level of HIV infection risk was considered as very high if the patient had previously used PEP and/or practiced chemsex and/or had an STI at initiation.

Assessment of the course of care during the first 12 months included attendance at one-on-one sessions of therapeutic support and counseling offered in our center, initial PrEP dosing, number of visits attended, additional visits for symptomatic STIs or side effects (besides the usual planned follow-up), follow-up status after last visit and retention rates in our center at 3, 6 and 12 months. Retention rate corresponded to the number of patients still followed in our center and not lost to follow-up out of the total number of patients included. Patients who stopped PrEP follow-up in our center (due to cessation of PrEP intake or continuation of follow-up elsewhere) were not considered as retained.

To assess the effectiveness of our program, we used the PrEP success ratio at 12 months proposed by Hendrickson et al., adapted to our local context [7]. PrEP follow-up was considered successful if the three following criteria were met: (1) the patient remained seronegative throughout follow-up, (2) the PrEP prescription was adapted to actual risk (including discontinuation if there was no indication), and (3) the patient was not lost to follow-up and missed at most one visit. Visits were considered missed if they took place more than 30 days after the expected date.

We performed descriptive analyses of the sociodemographic characteristics and sexual practices of patients at initiation, and of courses of care during the first 12 months of follow-up. All variables were described and expressed as frequency and percentage for categorical variables or median, minimum, maximum and interquartile range for continuous variables. Effectiveness of PrEP was assessed using the previously detailed success ratio and a 95% confidence interval was computed. We also described patients with unsuccessful PrEP follow-up. All analyses were performed using STATA/SE 16.0 (Stata Corporation, College Station, TX, USA).

Use of patients’ data for research received institutional review board approval from the French Data Protection Authority (authorization CNIL n° 1,320,749 v0). Patients were informed orally and by posters in the sexual health center of their right to object to the use of their data for research purposes. This procedure complies with French law, as observational studies on previously collected data require individual information and absence of patient objection (no explicit consent required). The Comité d’Evaluation de l’Ethique des projets de Recherche Biomédicale (CEERB) Paris Nord (Institutional Review Board -IRB 00006477- of HUPNVS, Paris 7 University, AP-HP), reviewed and approved the research project (n° CER-2020-38). The study was also declared to the AP-HP (Paris hospitals) Data Protection Office (n° 20200818150718).

## 3. Results

Between August 2018 and June 2019, 125 MSM initiated a first PrEP course in our center and were included in this study. They were mainly between 20 and 39 years old (*n* = 88, 71%) with a median age of 33, but the ages ranged up to 70 (Table 1). Almost three-quarters of the patients were born in France and lived in Paris. Most were employed (*n* = 102, 82%). Regarding sexual practices, more than 90% of patients had only male partners. Two-thirds had a history of STIs and 21% (*n* = 26) had an STI at initiation, mostly gonococcal infections (*n* = 13) and chlamydia (*n* = 8). More than half of patients (*n* = 73, 58%) were considered at very high risk of HIV infection, mainly due to a history of PEP.

At initiation, 77% of patients received one-on-one counseling and support on PrEP use (Table 2). The total number of counseling sessions ranged from zero to seven (median 1, Q1–Q3 1–2). Patients chose daily and on-demand PrEP dosing in equal proportions. During the first 12 months of follow-up, the total number of visits attended ranged from one to nine (median 3, Q1–Q3 2–4). Almost 20% of patients attended at least one additional visit besides the usual planned follow-up. Eleven patients decided to stop PrEP and four were followed in another center. More than 80% of patients chose quarterly follow-up in primary care with yearly visits in our sexual health center. Among them, 79% (*n* = 80) attended at least three visits in our center, and 48% (*n* = 48) attended at least two counseling sessions. After 12 months, the retention rate in our center was 84% (105/125).

Based on the PrEP success ratio at 12 months, 96% of patients (*n* = 120, 95% CI, 92.6–99.4) had a successful PrEP course. The five unsuccessful follow-ups were due to discontinuation by the patient. No HIV seroconversion was observed. Four of these five patients were younger than 40, two were born abroad, two lived outside Paris and all were employed or training. None of the five patients was considered at very high risk of HIV infection. They attended one to three visits before follow-up discontinuation.

## 4. Discussion

With the PrEP success ratio reaching 96% at 12 months, our results show that it is possible to achieve highly effective PrEP with no reported seroconversion in a real-world setting. This novel metric proposed by Hendrickson et al. offers a more appropriate assessment of PrEP success as it includes seroconversion, appropriateness of prescription according to HIV infection risk and regularity of follow-up [7].

Although we could not assess factors associated with PrEP success, all unsuccessful follow-ups (*n* = 5) occurred in patients who were not at very high risk of HIV infection. We could not collect the reasons for discontinuations of these five patients, but they could be explained by borderline indications at initiation in patients who did not have clear high-risk practices and were therefore more likely to drop PrEP. Although this indicator did not include all risk factors (notably the number of partners in the previous months), it highlighted the variety of PrEP indications and the absence of a clear risk threshold. This finding seems concordant with studies reporting low risk as a motive for discontinuing PrEP follow-up [8,9,10]. In total, 20 patients were not considered as retained in PrEP, including 15 patients who stopped PrEP or were followed elsewhere, and the 5 patients with an unsuccessful follow-up.

To our knowledge, this is the first study using the success ratio proposed by Hendrickson et al., which offers an adequate but still easy to use follow-up assessment metric. To compare with previous studies, we also assessed retention in PrEP. We found higher rates compared with previously published results regarding real-world settings. While retained participants at 12 months ranged from 28 to 60% in other publications, in our center, the retention rate at 12 months reached 84% [4,5,6,11,12]. The same results were observed at 3 months (56–87% vs. 90%) and 6 months (38–73% vs. 86%) [5,12,13,14,15,16,17,18,19,20,21,22,23,24]. Similar results were observed when trials and demonstration projects were included: retention rates expanded on wider ranges, but pooled estimates were still lower than our results (63% at 6 months and 71% at 12 months) [25]. The higher level of retention observed in our center compared with previous studies did not appear to be explained by differences in participants’ characteristics. As in other studies, patients were generally younger than 40 (between 62% and 81% in comparison with 71% in our patients) and were almost all MSM with only male sexual partners [2,4,5,6,11,19,20,21,22,26,27]. Characteristics regarding country of birth and work status were also similar in the studies for which this information was available [22,26,27]. Patients in our study showed a high (67%) but not unusual level of STI history; rates reported elsewhere ranged from 31% to 76% [2,17,22,24,26,27,28,29,30,31]. Studies also reported similar levels of STIs at initiation (21% vs. 17% to 26%) [12,21,24,27]. A history of PEP was also frequent although variable, at a rate of between 8% and 53% in comparison to 38% in our study [2,6,20,22,26,27,28,32]. Although direct and complete comparison is not possible, our population appeared to show a similar risk level for HIV infection as participants in previous studies.

The higher rate of retention that we report could be linked to the national and local setting. One important barrier to PrEP access and engagement identified in previous studies, especially in the USA, is its high cost [33,34]. The provision of visits, biological tests and antiretrovirals free of charge as part of the French PrEP prevention strategy may therefore facilitate patient follow-up. However, the lower rate of retention at 3 months reported in another French study (66% vs. 90% in our study) shows that besides financial barriers, other factors influence patients’ engagement in care [15]. Besides France, other countries offer access to PrEP free of charge or at reduced costs on a national scale or in particular settings [35]. Yet studies on PrEP retention in these countries also showed lower rates [22,26,27]. The particular setting of our sexual health center may therefore also play a part in the effectiveness of our PrEP program. Accompanying measures are offered through an identified and easily accessible referent counselor, including one-on-one sessions of therapeutic support and counseling. It has been previously shown that such sessions are considered as important facilitators of PrEP adherence [36,37].

In addition, qualitative studies have reported difficulties of scheduling and communication between providers and patients. Notably, the lack of a responsive and identified PrEP navigator as well as the need for more adherence counseling and education information were highlighted [33,38]. Support from a counselor could also be useful to facilitate transition from PEP to PrEP, which was associated with improved PrEP persistence [30]. The accompanying measures set up in our center may thus play a key role in maintaining care engagement and improving follow-up quality.

### Limitations

This study has some limitations. First, it was conducted on a limited number of patients. Because of the high level of PrEP success, we only observed five unsuccessful follow-ups and we were not able to analyze factors associated with PrEP success. Second, this study focused only on the first 12 months of follow-up. While it gave us a reliable overview of the follow-up at its beginning, it did not take into account the long-term evolution of PrEP user engagement. Third, our study did not allow us to assess the specific impact of a quarterly follow-up in primary care, notably regarding the accompanying measures. Such a follow-up could have both positives (easier access to a family physician) and negatives (reduced access to information and counseling in our sexual health center) consequences on PrEP success. However, in our setting, accompanying measures and in particular access to the referent were still available and used by patients followed in primary care. Moreover, until now, the few studies on PrEP retention in primary care have shown mediocre retention rates, notably among family physicians less experienced with PrEP [19,24,39]. Finally, these results cannot be generalized to all PrEP users as we only included MSM participants. Although PrEP is recommended in all populations at high risk of HIV infection, so far it has been mostly prescribed in MSM, and most of the studies regarding PrEP follow-up have been conducted among this population.

## 5. Conclusions

This study highlights the possibility of achieving a high PrEP success ratio among MSM in a real-world setting such as in a sexual health center. Although associated factors could not be analyzed, it can be hypothesized that our specific setting relying on accompanying measures and one-on-one counseling by a trained counselor could explain the improved effectiveness of our PrEP program. These results strengthen the body of evidence regarding the benefits of such interventions, but they need to be confirmed through comparisons with other centers, especially to clarify the potential effects of primary care follow-up.

## Figures and Tables

**Table 1 ijerph-19-14295-t001:** Sociodemographic characteristics and sexual practices of patients using HIV pre-exposure prophylaxis at the first consultation.

	Frequency (*n* = 125)	Percent
**Age (years)**		
20–29	47	38
30–39	41	33
40–49	24	19
≥50	13	13
**Born in France**		
Yes	91	73
No	34	27
**French nationality**		
Yes	80	64
No	26	21
Missing	19	15
**Work status**		
Employed	102	82
Training	12	10
No professional activity	10	8
Missing	1	1
**Place of residence**		
Paris	95	76
Outside Paris	30	24
**Sexual partners**		
Men	116	93
Men and women	9	7
**Chemsex practice**		
Yes	21	17
No	97	78
Missing	7	6
**History of STIs**		
Yes	84	67
No	34	27
Missing	7	6
**History of PEP**		
Yes	48	38
No	75	60
Missing	2	2
**STI at initiation**		
Yes	26	21
No	99	79
**Risk of HIV infection considered as very high ^1^**		
Yes	73	58
No	52	42

^1^ includes chemsex practice, STI at initiation and/or history of PEP.

**Table 2 ijerph-19-14295-t002:** Course of care and successful HIV pre-exposure prophylaxis at 12 months.

	Frequency (*n* = 125)	Percent
**One-on-one counseling and support at initiation**		
Yes	96	77
No	29	23
**Initial PrEP dosing**		
Daily	61	49
On demand	64	51
**Additional visits during the first 12 months**		
Yes	24	19
No	101	81
**Follow-up status after the last visit**		
PrEP stopped	11	9
Followed elsewhere	4	3
Quarterly follow-up in local center	9	7
Quarterly follow-up in general practice	101	81
**Retention in the local center at 3 months**		
Yes	112	90
No	13	10
**Retention in the local center at 6 months**		
Yes	108	86
No	17	14
**Retention in the local center at 12 months**		
Yes	105	84
No	20	16
**PrEP success at 12 months**		
Yes	120	96
No	5	4

## Data Availability

Data are available on request due to restrictions, e.g., privacy or ethical.

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
