# Peer review of "Is HIV Pre-Exposure Prophylaxis among Men Who Have Sex with Men Effective in a Real-World Setting? Experience with One-On-One Counseling and Support in a Sexual Health Center in Paris, 2018–2020"

_ijerph, 2022, doi:10.3390/ijerph192114295_

Round 1

Reviewer 1 Report

1.     The citations (numbers) should be provided at the end of the sentence – please correct it in all manuscript

2.     Line 82 “Patients who stopped PrEP or were followed elsewhere were not considered as retained. “ – please provide clear exclusion and inclusion criteria

3.     The limitations of the study should be provided at the end of discussion section

4.     The statistical analyses are poor, you mentioned that All analyses were per- 94 formed using STATA/SE 16.0  - could you provide the names of the statistical analyses that you performed?

5.     Please develop the discussion section providing more studies to compare with your study

6.     In the line 111 you mentioned: Two-thirds 111 had a history of STI and 21% (n = 26) had an STI at initiation, mostly gonococcal infections 112 (n = 13) and chlamydia (n = 8). More than half of patients (n = 73, 58%) were considered at 113 very high risk of HIV infection, mainly due to a history of PEP. 

Could you explain how the STIs were diagnosed among these patients?

Author Response

Dear reviewer,
Thank you for your review of our manuscript regarding PrEP effectiveness in our center. Here are our responds to the comments you made:

  1. The citations (numbers) should be provided at the end of the sentence – please correct it in all manuscript

Citations position has been corrected in line 51, 87, 146, 164, 198 and 200.

  1. Line 82 “Patients who stopped PrEP or were followed elsewhere were not considered as retained.“ – please provide clear exclusion and inclusion criteria

This sentence does not refer to inclusion or exclusion criteria (listed lines 66-67) but to the definition of retention in our study, as it’s usually employed in PrEP effectiveness studies. We considered retained all patients who were still followed for PrEP in our center after 12 months. All the other patients were not considered as retained. This includes patients who decided to stop PrEP intake (often because they are no longer exposed to at-risk situations), so they no longer require follow-up, as well as those who continue their follow-up in another center (for instance after moving). We changed sentence lines 84-85 to clarify this definition.

“Patients who stopped PrEP or were followed elsewhere were not considered as retained.”

-> “Patients who stopped PrEP follow-up in our center (due to cessation of PrEP intake or continuation of follow-up elsewhere) were not considered as retained.”

  1. The limitations of the study should be provided at the end of discussion section

Limitations have been moved at the end of discussion section (lines 203-220).

  1. The statistical analyses are poor, you mentioned that All analyses were per- 94 formed using STATA/SE 16.0 - could you provide the names of the statistical analyses that you performed?

We performed descriptive analyses, expressed as frequency and percentage for categorical variables or median, minimum, maximum and interquartile range for continuous variables. Effectiveness of PrEP was assessed using the success ratio (see lines 86-92) and 95% confidence interval was computed.

Modifications were made lines 95-98 to add these precisions.

“We performed descriptive analyses of sociodemographic characteristics and sexual practices of patients at initiation, and of courses of care during the first 12 months of follow-up. Effectiveness of PrEP was assessed using the previously detailed success ratio. We also described patients with un-successful PrEP follow-up. All analyses were performed using STATA/SE 16.0 (Stata Corporation, College Station, TX, USA).”

-> “We performed descriptive analyses of sociodemographic characteristics and sexual practices of patients at initiation, and of courses of care during the first 12 months of follow-up. All variables were described and expressed as frequency and percentage for categorical variables or median, minimum, maximum and interquartile range for continuous variables. Effectiveness of PrEP was assessed using the previously detailed success ratio and 95% confidence interval was computed. We also described patients with un-successful PrEP follow-up. All analyses were performed using STATA/SE 16.0 (Stata Corporation, College Station, TX, USA).”

  1. Please develop the discussion section providing more studies to compare with your study

We have developed the discussion section to improve comparison with others recent studies (published between 2018 and 2022), regarding characteristics of participants as well as retention rates. We have also added references regarding the potential impact of costs as well as primary care follow-up on PrEP retention.

Modifications were made lines 162 to 177, 186 to 189 and 209 to 217. Added references are listed below:

Flores Anato, J.L.; Panagiotoglou, D.; Greenwald, Z.R.; Trottier, C.; Vaziri, M.; Thomas, R.; Maheu-Giroux, M. Chemsex Practices and Pre-Exposure Prophylaxis (PrEP) Trajectories among Individuals Consulting for PrEP at a Large Sexual Health Clinic in Montréal, Canada (2013-2020). Drug and Alcohol Dependence 2021, 226, 108875, doi:10.1016/j.drugalcdep.2021.108875.

Hevey, M.A.; Walsh, J.L.; Petroll, A.E. PrEP Continuation, HIV and STI Testing Rates, and Delivery of Preventive Care in a Clinic-Based Cohort. AIDS Education and Prevention 2018, 30, 393–405, doi:10.1521/aeap.2018.30.5.393.

Rotsaert, A.; Reyniers, T.; Jacobs, B.K.M.; Vanbaelen, T.; Burm, C.; Kenyon, C.; Vuylsteke, B.; Florence, E. PrEP User Profiles, Dynamics of PrEP Use and Follow‐up: A Cohort Analysis at a Belgian HIV Centre (2017–2020). J Int AIDS Soc. 2022, 25, doi:10.1002/jia2.25953

Bruxvoort, K.J.; Schumacher, C.M.; Towner, W.; Jones, J.; Contreras, R.; Ling Grant, D.; Hechter, R.C. Referral Linkage to Preexposure Prophylaxis Care and Persistence on Preexposure Prophylaxis in an Integrated Health Care System. JAIDS Journal of Acquired Immune Deficiency Syndromes 2021, 87, 918–927, doi:10.1097/QAI.0000000000002668.

Havens, J.P.; Scarsi, K.K.; Sayles, H.; Klepser, D.G.; Swindells, S.; Bares, S.H. Acceptability and Feasibility of a Pharmacist-Led Human Immunodeficiency Virus Pre-Exposure Prophylaxis Program in the Midwestern United States. Open Forum Infectious Diseases 2019, 6, ofz365, doi:10.1093/ofid/ofz365.

Stankevitz, K.; Grant, H.; Lloyd, J.; Gomez, G.B.; Kripke, K.; Torjesen, K.; Ong, J.J.; Terris-Prestholt, F. Oral Preexposure Prophylaxis Continuation, Measurement and Reporting. AIDS 2020, 34, 1801–1811, doi:10.1097/QAD.0000000000002598.

Uhrmacher, M.; Skaletz-Rorowski, A.; Nambiar, S.; Schmidt, A.J.; Ahaus, P.; Serova, K.; Mordhorst, I.; Kayser, A.; Wach, J.; Tiemann, C.; et al. HIV Pre-Exposure Prophylaxis during the SARS-CoV-2 Pandemic: Results from a Prospective Observational Study in Germany. Front. Public Health 2022, 10, 930208, doi:10.3389/fpubh.2022.930208.

Hovaguimian, F.; Martin, E.; Reinacher, M.; Rasi, M.; Schmidt, A.; Bernasconi, E.; Boffi El Amari, E.; Braun, D.; Calmy, A.; Darling, K.; et al. Participation, Retention and Uptake in a Multicentre Pre‐exposure Prophylaxis Cohort Using Online, Smartphone‐compatible Data Collection. HIV Medicine 2022, 23, 146–158, doi:10.1111/hiv.13175.

Serota, D.P.; Rosenberg, E.S.; Sullivan, P.S.; Thorne, A.L.; Rolle, C.-P.M.; Del Rio, C.; Cutro, S.; Luisi, N.; Siegler, A.J.; Sanchez, T.H.; et al. Pre-Exposure Prophylaxis Uptake and Discontinuation Among Young Black Men Who Have Sex With Men in Atlanta, Georgia: A Prospective Cohort Study. Clinical Infectious Diseases 2020, 71, 574–582, doi:10.1093/cid/ciz894.

Spinelli, M.A.; Scott, H.M.; Vittinghoff, E.; Liu, A.Y.; Gonzalez, R.; Morehead-Gee, A.; Gandhi, M.; Buchbinder, S.P. Missed Visits Associated With Future Preexposure Prophylaxis (PrEP) Discontinuation Among PrEP Users in a Municipal Primary Care Health Network. Open Forum Infectious Diseases 2019, 6, doi:10.1093/ofid/ofz101.

Hayes, R.; Schmidt, A.J.; Pharris, A.; Azad, Y.; Brown, A.E.; Weatherburn, P.; Hickson, F.; Delpech, V.; Noori, T.; the ECDC Dublin Declaration Monitoring Network Estimating the ‘PrEP Gap’: How Implementation and Access to PrEP Differ between Countries in Europe and Central Asia in 2019. Eurosurveillance 2019, 24, doi:10.2807/1560-7917.ES.2019.24.41.1900598.

Garrison, L.E.; Haberer, J.E. Pre-Exposure Prophylaxis Uptake, Adherence, and Persistence: A Narrative Review of Interventions in the U.S. American Journal of Preventive Medicine 2021, 61, S73–S86, doi:10.1016/j.amepre.2021.04.036.

Chidwick, K.; Pollack, A.; Busingye, D.; Norman, S.; Grulich, A.; Bavinton, B.; Guy, R.; Medland, N. Utilisation of Pre-Exposure Prophylaxis (PrEP) for HIV Prevention in the Australian General Practice Setting: A Longitudinal Observational Study. Sex. Health 2022, doi:10.1071/SH21207.

Velloza, J.; Donnell, D.; Hosek, S.; Anderson, P.L.; Chirenje, Z.M.; Mgodi, N.; Bekker, L.-G.; Marzinke, M.A.; Delany-Moretlwe, S.; Celum, C. Alignment of PrEP Adherence with Periods of HIV Risk among Adolescent Girls and Young Women in South Africa and Zimbabwe: A Secondary Analysis of the HPTN 082 Randomised Controlled Trial. The Lancet HIV 2022, 9, e680–e689, doi:10.1016/S2352-3018(22)00195-3.

Vanbaelen, T.; Rotsaert, A.; Jacobs, B.K.M.; Florence, E.; Kenyon, C.; Vuylsteke, B.; Laga, M.; Thijs, R. Why Do HIV Pre-Exposure Prophylaxis Users Discontinue Pre-Exposure Prophylaxis Care? A Mixed Methods Survey in a Pre-Exposure Prophylaxis Clinic in Belgium. AIDS Patient Care and STDs 2022, 36, 159–167, doi:10.1089/apc.2021.0197.

  1. In the line 111 you mentioned: Two-thirds 111 had a history of STI and 21% (n = 26) had an STI at initiation, mostly gonococcal infections 112 (n = 13) and chlamydia (n = 8). More than half of patients (n = 73, 58%) were considered at 113 very high risk of HIV infection, mainly due to a history of PEP. Could you explain how the STIs were diagnosed among these patients?

Following national and international recommendations, systematic screening of STIs were realized before initiation of PrEP. This screening includes serologies on blood sample for HIV and syphilis, and nucleic acid amplification testing on urine samples, pharyngeal and rectal self-swabs for chlamydia and gonococci and other STIs. Precisions were added in the methods section, lines 73-75.

“Routinely collected data included patients’ sociodemographic characteristics (age, nationality, place of birth and residence, work status), sexual practices (partners’ gender, history of STI, PEP and chemsex practice in the last 12 months) and STI at initiation (based on test including HIV and other STIs).”

-> “Routinely collected data included patients’ sociodemographic characteristics (age, nationality, place of birth and residence, work status), sexual practices (partners’ gender, history of STI, PEP and chemsex practice in the last 12 months) and STI at initiation based on systematical biological tests performed before initiation. Tests results included serologies on blood sample for HIV and syphilis, and nucleic acid amplification testing on urine samples, pharyngeal and rectal self-swabs for chlamydia and gonococci.”

We hope that these information and modifications answer your requests and questions.

Sincerely,

The authors

Reviewer 2 Report

The study entitled “Is HIV pre-exposure prophylaxis among men who have sex with men effective in a real-world setting? Experience with one-to-one counseling and support in a sexual health center in Paris, 2018-2020” presents some weaknesses:

This is a retrospective study with a follow-up of only 12 months. The authors only describe the characteristics of the patients. According to the authors, this study aimed to assess the effectiveness of their PreP program that included accompanying measures with a counselor. However, this objective has not been achieved since they have not been able to collect reasons for the discontinuation of PreP. On the other hand, we do not know if these accompanying measures have also been followed in primary care. This data is essential since 81% of the patients performed the quarterly follow-up in general practice. The effects of this follow-up in general practice have not been able to be measured. The patients who were followed up in general medicine had only two visits to the sexual health center. Maybe one at the beginning and another at the end of the 12 months. Therefore, it is essential to know what happens in primary care consultations before attributing the high percentage of retention to the influence of the counselor.

Author Response

Dear reviewer,
Thank you for your review of our manuscript regarding PrEP effectiveness in our center. Here are our responds to the comments you made:

  1. Non-collection of discontinuation reasons

Among our study population of 125 patients, we had:

  • n=105 who remained followed in the PrEP program,
  • n=11 who stopped PrEP intake as they were no longer exposed to at-risk situations,
  • n=4 who were followed in another center,
  • n=5 who were lost to follow-up without any information on reasons of discontinuation.

In total, among the 20 patients who were not retained in our local center at 12 months, we were indeed able to collect discontinuation reasons for only 15 patients at the end of the last consultation or after through exchanges with the referent counselor.

Thus, and unlike studies using retention in PrEP to assess follow-up quality, we were able to categorize these 15 courses of care as successful according to the success ratio developed by Hendrickson et al.

To clarify this, we’ve added precisions in the discussion section, lines 148-149 and 155-157.

“Although we could not assess factors associated with PrEP success, all unsuccessful follow-ups occurred in patients who were not at very high risk of HIV infection. We could not collect reasons for discontinuations, but they could be explained by borderline indications at initiation […] This finding seems concordant with studies reporting low risk as a motive for discontinuing PrEP follow-up.”

-> “Although we could not assess factors associated with PrEP success, all unsuccessful follow-ups (n=5) occurred in patients who were not at very high risk of HIV infection. We could not collect reasons for discontinuations of these 5 patients, but they could be explained by borderline indications at initiation […] This finding seems concordant with studies reporting low risk as a motive for discontinuing PrEP follow-up. In total, 20 patients were not considered as retained in PrEP, including 15 patients who stopped PrEP or were followed elsewhere, and the 5 patients with an unsuccessful follow-up.”

  1. Accessibility of accompanying measures in primary care

Regarding the role of accompanying measures for patients followed in primary care, two comments can be made:

First, part of the follow-up was made in our center for these patients. As a matter of fact, more than three quarters of them (79%, n=80) attended at least 3 visits in our center, and almost half of them (48%, n=48) attended at least two sessions of one-on-one counseling and support.

Second, accompanying measures were not limited to the counseling sessions. It also included support in organizing appointments with reminders, as well as phone and email follow-up to answer daily questions. This help was available and accessible for all patients followed in our center, including those with quarterly follow-up in primary care. Although collected data did not enable us to quantify this help, our referent counselor did actually provide this kind of support for some of those patients.

To better describe exposition to accompanying measures, precisions on the results have been added lines 126-128 and 131-132, and the discussion section was developed as well (lines 213-215)

“At initiation, 77% of patients received one-on-one counseling and support on PrEP use (Table 2). […] More than 80% of patients chose quarterly follow-up in primary care with yearly visits in our sexual health center.”

-> “At initiation, 77% of patients received one-on-one counseling and support on PrEP use (Table 2). Total number of counseling sessions ranged from 0 to 7 (median 1, Q1-Q3 1-2). […] More than 80% of patients chose quarterly follow-up in primary care with yearly visits in our sexual health center. Among them, 79% (n=80) attended at least three visits in our center, and 48% (n=48) attended at least two counseling sessions.”

  1. Measurement of effect of general medicine follow-up

Regarding the general medicine follow-up, it could indeed influence PrEP retention, notably through easier access to one’s family physician. While we hope primary care follow-up will eventually have a positive impact, the quality of follow-up has not necessarily improved until now, as shown by some studies conducted in primary care settings. Retention rates are still mediocre in such settings, notably among physicians less experienced with PrEP (see Chidwick 2022, Havens 2019 and Spinelli 2019). The potential barriers and difficulties described for primary care follow-up are the lack of awareness and training about PrEP, as well as difficulty talking freely about sex life, especially to physician who are not LGBTQ + affirming (see Chiarabini 2021, DOI: 10.3917/spub.211.0101; Storholm 2021, DOI: 10.1521/aeap.2021.33.4.325; and Turner 2018, DOI: 10.1016/j.jana.2017.11.002).

To highlight this, the discussion section has been developed to include these limitations (lines 209 to 217) and references cited above have been added. As the effects of such limitations need to be explored in future studies, we have nuanced the conclusion (lines 227-229).

We hope that these information and modifications answer your requests and questions.

Sincerely,

The authors

Round 2

Reviewer 2 Report

I have read the revised version of the article. The authors make a retrospective and descriptive study of their experience with their PrEP program. I think that with the available data, the effectiveness of their PrEP program cannot be definitively attributed to the accompanying measures, although it does allow them to think of it as a hypothesis.

Otherwise, the authors have answered the questions raised in the previous review.

Author Response

We thank you for your new feedback, we have revised the conclusion in order to emphasize that it is only a hypothesis (lines 223-226).

"Although associated factors could not be analyzed, it seems that our specific setting relying on accompanying measures and one-on-one counseling by a trained counselor resulted in improved effectiveness of our PrEP program."

-> "Although associated factors could not be analyzed, it can be hypothesized that our specific setting relying on accompanying measures and one-on-one counseling by a trained counselor could explain the improved effectiveness of our PrEP program."

Sincerely,

The authors